# Genetic and Environmental Contributors to Neurodegeneration: An Exploration of the Effects of Alcohol on Clinical Features of Huntington’s Disease Using the Enroll-HD Global Platform

**DOI:** 10.3390/ijerph18105113

**Published:** 2021-05-12

**Authors:** Alexander L. Symonds, Antonella Macerollo, Kevin Foy, Sundus H. Alusi, Rhys Davies

**Affiliations:** Department of Neurology, The Walton Centre NHS Foundation Trust, Liverpool L9 7LJ, UK; symondsal@gmail.com (A.L.S.); A.Macerollo@liverpool.ac.uk (A.M.); Kevin.Foy@thewaltoncentre.nhs.uk (K.F.); Sundus.Alusi@thewaltoncentre.nhs.uk (S.H.A.)

**Keywords:** Huntington’s disease, alcohol, neurodegeneration

## Abstract

Huntington’s disease (HD) is a neurodegenerative dementia with a well recognised genetic cause. Alcohol misuse is a major environmental factor relevant to numerous neurological presentations, including HD. We explored the effects of alcohol intake on clinical features of HD by means of data from the Enroll-HD, which is a global registry study. A retrospective observational study making use of the Enroll-HD periodic dataset up to 2020 (in accordance with the Enroll-HD guidelines, encompassing 16,120 subjects with the HD gene (CAG expansion > 36), was carried out. This included 180 sites in 21 countries. The study looked at the association of alcohol use with the clinical presentation of HD, specifically looking into the age of first symptoms and HD severity. We also describe a specific case with manifest HD, a participant in the Enroll-HD study, whereby the patient’s obsessionality was central to her pattern of high alcohol intake and to her successful avoidance of alcohol thereafter. A record of past problems with high alcohol intake was more common in the group with manifest HD (9.0%, *n* = 1121) when compared with the pre-manifest carriers of the HD genetic abnormality (2.3%, *n* = 339). Age at onset of symptoms was not significantly influenced by current alcohol misuse, or past misuse. The severity of clinical impairments in HD was influenced by alcohol. Patients who reported high alcohol intake in the past had a statistically significant increase in motor impairments, by the Unified Huntington’s Disease Rating Scale total motor score (Kruskal–Wallis, post hoc Dunn’s, *p* < 0.001), and a significantly higher burden of psychiatric symptoms by the Problem Behaviours Assessment score (Kruskal–Wallis, post hoc Dunn’s, *p* < 0.01) compared with those not reporting high alcohol use. However, the past alcohol group did not have a lower Mini Mental State Examination score (Kruskal–Wallis, post hoc Dunn’s, *p* > 0.05) The first symptom of HD, as determined by the assessing clinician, was more likely to be psychiatric disturbance in patients currently misusing alcohol or those with prior history of alcohol misuse (55% and 31% respectively) when compared with controls (5%). Individual case experience, such as that presented in this study, shows that HD and alcohol, two major genetic and environmental contributors to neurodegeneration, interact in producing clinical problems. However, the complexities of these interactions are difficult to define, and may require larger studies dedicated to exploring the various factors in this interaction.

## 1. Introduction

Huntington’s disease (HD) is a neurodegenerative disorder characterised by a triad of abnormal movements, psychiatric symptoms and progressive cognitive impairment or dementia [1]. Currently, the evidence for an effect of alcohol on HD presentation and disease course is limited. Psychiatric and behavioural changes are the most common initial symptoms in HD, with depression occurring in half of the population [2]. Irritability, aggression and apathy are common features, with apathy being the best neuropsychiatric predictor of disease progression [3]. In addition to depression, psychosis and obsessive-compulsive symptoms can also be present [4,5]. There is increasing evidence for the role of alcohol consumption on HD onset [6]. However, it is uncertain whether alcohol dependence is more common in the HD population [7]. A number of studies have suggested a link between HD and higher prevalence of alcohol abuse [8,9,10,11], although Pflanz et al. suggested a similar rates to the population [12]. In addition to rates of alcohol abuse, King proposed that most drinkers started abusing alcohol before manifesting symptoms [13]. The relationship between ongoing alcohol misuse or of prior alcohol misuse and symptom onset or course is still unclear. Psychiatric symptoms such as depression have previously been shown to be more common in those patients who misuse alcohol [13]. 

Both acute and chronic high intake of alcohol use can contribute to motor dysfunction and a decline in cognition in general [14]. Chronic alcohol use can lead to alcohol related brain injury and dementia as well as the more defined Wernicke–Korsakoff’s syndrome [14].

The burden of alcohol misuse in patients with neurodegenerative dementia can accentuate cognitive decline and in some studies has been shown to affect the initial clinical presentation [15]. Our study looks into the influence both of past and current, ongoing high alcohol intake use the clinical phenotype of patients with HD. In doing so, we explored both a well-recognised environmental contributor to dementia syndromes and a well-defined monogenetic factor contributing to neurodegenerative cognitive impairment. 

## 2. Methods

Enroll-HD is an international observational study of HD with more than 16,000 subjects at 180 sites in 21 countries. Most participants in Enroll-HD are diagnosed with HD but others are subjects identified as having the HD gene expansion that leads to symptomatic HD. The database also includes a minority of HD family members who are genetically untested and other contacts, but these participants did not contribute to our analyses.

On the initial visit, participants are asked about medical and family histories as well as substance use. Follow up visits record progression of motor, cognitive and psychiatric symptoms. The presented study was a retrospective observational study, making use of the Enroll-HD periodic dataset up to 2020 (in accordance with the Enroll-HD guidelines https://www.enroll-hd.org/ accessed on 28 January 2021). This anonymized dataset was stored in Microsoft Excel, Microsoft Corporation, Seattle, United States and subsequently analysed in GraphPad Prism, GraphPad Software incorporated, California, United States.

Estimated age at onset of symptoms was determined by the clinician from the clinical history. The profile of HD symptoms was documented by means of validated instruments, in accordance with Enroll-HD procedures. The Unified Huntington’s Disease Ratings Scale (UHDRS) total motor score was used for motor features [16], and the Mini Mental State Examination (MMSE) for cognition [17]. The Problem Behaviours Assessment (PBA) and its five subscores (depression, irritability, psychosis, apathy and executive function) provided a quantitative measure for psychiatric burden [18]. The neuropsychiatric profile of HD was further determined by referring to symptom onset data as determined by the clinician.

Alcohol use was recorded as part of each Enroll-HD visit with participants being asked if there was a history of ‘alcohol problems in the past’ and asked about current alcohol consumption. Participants were grouped into controls (below 14 units a week), current high users (greater than 30 units a week) and those with a history of alcohol problems (‘alcohol problems in the past’). Participants whose data did not conform to these distinct categories were not included in the analyses. The inclusion and exclusion of patients is summarised in the Figure 1. Enroll-HD participants sign an informed consent at recruitment. All sites followed local ethical guidelines and had local ethical committee approval for the recording of patient data. 

An illustrative case showing a distinctive relationship between alcohol and clinical manifestations of HD is also described. 

## 3. Results 

### 3.1. Demographics

The Enroll-HD periodic dataset included 16,120 HD patients from Europe, the Americas and Australasia; the majority were caucasian (*n* = 15,102). Patients were split 7393:8727 male to female; the mean education level (ISCED) was 3.6. ISCED level 3 is equivalent to the completion of secondary education. The majority of HD patients presenting to Enroll-HD clinics had manifest disease; 5173 were pre-manifest. Regardless of the earliest symptom type, the onset of symptoms was at 45.2 years old, as judged by the clinician. In comparison, the first visit was on average 3.5 years later (48.7 years), again implying that more patients with manifest disease, than pre-manifest, were under follow up

The most frequently designated HD symptom defining onset of manifest HD was motor (as defined by UHDRS, usually involuntary movement in the form of chorea, 52%). The frequency of mixed (19%), psychiatric (21%) or cognitive (8%) symptoms as defining the onset of manifest disease was lower. However, mean ages for the earliest symptoms in the three categories of motor, cognitive and psychiatric were 59, 43, 50 years, respectively, implying that clinicians have greater confidence in attributing motor symptoms to HD itself while other features are more likely to be formulated as incidental or, perhaps, prodromal.

### 3.2. The Effect of Alcohol on Onset of Huntington’s Disease (HD)

The prevalence of current high alcohol use (greater than 30 units per week) in the combined manifest and premanifest HD population was relatively small (3%, *n* = 222). However, a much larger number of subjects reported having a problem with alcohol use in the past (9%, *n* = 1460). A history of alcohol problems in the past was more common in the manifest HD group (7.7%, *n* = 1121) when compared with the premanifest (2.3%, *n* = 339). The overlap between those patients who currently had high alcohol use and those who had previously reported having alcohol problems was small. The number of patients who were both using large amounts of alcohol at the time of assessment and reported alcohol problems in the past was just 0.6% (*n* = 85).

Cumulatively, the frequency of recreational drug use within the HD population (12.0%, *n* = 1962) was high. This figure encompasses all recreational substances other than alcohol but is not readily subdivided by specific substance or prior/ongoing patterns of use. Interestingly, the number of subjects using recreational substances other than alcohol was split evenly between patients who had manifest symptoms (*n* = 1057) and those who are pre-manifest (*n* = 905). 

The age at onset of symptoms was similar in those who had a history of past alcohol use and those who consumed high levels of alcohol at the time of the data collection. Those who did not report drinking (average onset 43.2 years) had an onset of symptoms similar to that in subjects with past alcohol problems (43.1) but somewhat younger than those describing current high levels of alcohol consumption (45.6). Breaking down the age at onset of HD into its underlying core symptoms modalities (motor, cognitive or psychiatric), there was no correlation between past alcohol use and the age of onset of particular domain. 

The first symptom identified in HD by the clinician seemed to vary, depending on whether the patient has a history of alcohol misuse, or was currently misusing alcohol. The proportion of patients presenting with motor symptoms was similar between those who drink minimal amount of alcohol, those currently misusing alcohol and those who had misused alcohol in the past (37.2%, 41.2% and 40.2% respectively). However, those who are currently misusing alcohol, or had misused it in the past, were more likely to have a psychiatric symptom identified as their initial clinical feature first (55.3% and 30.7%) by comparison with those who do not drink excessively (4.9%). 

### 3.3. The Effect of Alcohol on HD Severity

In addition to age at onset of disease, severity of HD was also influenced by alcohol (Figure 1). The average total motor score of UHDRS in the control group was lower (27) compared with patients who reported alcohol problems in the past (29) and the difference was statistically significant (Kruskal–Wallis, post hoc Dunn’s, *p* < 0.001).

The MMSE tool was used as a simple screen for cognitive impairment, with scores of below 24 defining a dementia. [19]. By contrast to motor symptoms, patients who reported past problems with alcohol did not have a significantly lower MMSE score (26) compared with control group (26, Kruskal–Wallis, post hoc Dunn’s, *p* > 0.05, Figure 1). There was also no difference between those who used high levels of alcohol at the time of the study and the control group (27 and 26 respectively, Kruskal–Wallis, post hoc Dunn’s, *p* > 0.05).

As for the PBA components of depression, irritability, psychosis, apathy and executive function, patients who had a prior history of alcohol abuse had higher scores (indicating greater burden) in all these domains when compared with the control group. However, this was not the case for patients currently drinking excessively (Figure 2, Kruskal–Wallis, post hoc Dunn’s, *p* < 0.01).

### 3.4. Case Vignette

Our patient had a positive predictive genetic test for HD (CAG repeat lengths: >39, 32) at age 37, which was some years after her father had been diagnosed with the condition. Premorbidly, her personality had been quiet and shy. She lived with her husband and two children, both younger than 18 at the time of the predictive test.

Following her first neurology outpatient appointment at 45, she developed increasing impulsivity but also complusive behaviour and only mild motor impairment. Having previously never had a problematic relationship with alcohol use, she was described by her family as displaying stereotyped drinking behaviour that started in the early evening with beer and was followed by wine, white then red. They were unable to dissuade her from drinking in this way or from several other compulsive behaviours, which included habitual checking and extreme cleanliness.

Prior to her HD diagnosis, she only drank alcohol on social occasions, and very rarely consumed alcohol at home. There was no history of other recreational drug use or psychiatric illness. An escalation in confused aggression and impulsive behaviour and increasing risk of falls while intoxicated led to emergency admission for psychiatric assessment and treatment. Following this admission of several weeks, her routines changed. Alcohol was no longer the focus of her compulsions, but rather patterns of taking exercise. Her family noted a major, enduring improvement in HD impairments in all three domains of motor function, psychiatric disturbance and cognition.

## 4. Discussion

We describe a specific patient with HD for whom obsessional traits were key both to the development and resolution of alcohol misuse. Of course, psychological distress from the fact of a diagnosis of progressive neurological disease and from awareness of neurological impairment will have been additional contributors. In our patient, direct effects of alcohol intoxication severely compounded neuropsychiatric and motor symptoms from the genetically confirmed dementing neurodegenerative disease. Triggered by our experience of this case, we took the opportunity of interrogating the global Enroll-HD database for any relationship between alcohol intake, previous or current, and clinical features in HD.

While we found that alcohol misuse in HD as a problem in a significant minority of the HD population, specific patterns were difficult to elucidate despite the very clear categories used for the analysis and the very large sample at our disposal. The Enroll-HD data included substantially more patients who had a past history of problem alcohol use than those who were drinking heavily at the time of their Enroll HD assessment. Of the three domains of HD symptom, psychiatric symptoms occurred on average at the earliest age on average, followed by motor and then cognitive symptoms. As the diagnosis of manifest HD for practical purposes is made at the onset of motor symptoms, these early psychiatric features are inferred to have been prodromal, i.e., occurring before manifest HD is diagnosed [1]. For those patients who had a prior history of problem alcohol use, the age at onset of HD was not significantly earlier. Neither were those who were drinking greater than 30 units at time of assessment diagnosed with HD at a significantly earlier age. However, a potential limitation here is not knowing when the problem alcohol use occurred in relation to age of onset of symptoms.

By contrast with the lack of association of excessive alcohol intake on HD onset, HD severity profile, as measured by the UHDRS total motor score, was shown to be worse in those with documented previous problem alcohol use. Overall, patients with a history of alcohol use had a higher burden of impairment as measured by UHDRS and the PBAs, but not the MMSE (Figure 1). Although the MMSE is widely used as a dementia screening tool, its ability detect incremental changes in cognition is limited. Ongoing high levels of alcohol use did not, on the other hand, appear to correlate with increased burden or an earlier age of onset of HD. Of note, acute alcohol intoxication can cause many symptoms that could mimic those of HD such as poor attention and concentration, changes to mood, imbalance and falls. Our data do not show whether neurological impairment was a factor that may have led to curtailment of alcohol use. Alcohol appeared to influence some psychiatric symptoms more than others, specifically depression and irritability. Whether these symptoms are more common comorbidities or more common as a consequence of alcohol is not something that was looked at specifically in this study. The role of alcohol in psychiatric symptoms in particular is a well-known phenomenon, that appears to also apply to HD patients.

Within the PBA there are a number of psychiatric symptoms evaluated, including depression, irritability and apathy as mentioned above. Psychosis and other obsessional symptoms (described as executive function) are within the assessment. The increase in burden of symptoms is conserved across these domains and are not limited to common psychiatric complaints, such as depression. The overall increase in burden of each of these symptoms in patients who have historically abused alcohol, but not in those that currently abuse alcohol, was unexpected.

The correlation between a past problem alcohol use and the increase in the degree of impairment does not give a clear answer as to causation. A history of heavy alcohol use was more common in those patients who had manifest disease (1460 compared to 222). Could this perhaps suggest that patients are curbing their drinking to stop the deterioration of symptoms? Further investigation is needed to elucidate the contributing factors to patients’ cessation of high alcohol intake.

The main limitations to this study relate to the covariates to clinical HD presentation, including CAG repeat length, as well as the use of MMSE as a cognitive assessment score. Age, CAG repeat length, total motor score and aspects of cognitive assessment can all be used to help guide expected prognosis [20] but we were unable to include these co-variates in the analysis. Furthermore, the detail concerning alcohol intake in this study is subject to patient recall bias. In addition, high alcohol intake may be underrepresented in patients who chose to engage with a research project, such as Enroll-HD. In making use of a large dataset we are dependent on relatively simple measures, which is especially limiting in the case of using MMSE as our only cognitive measure.

The relationship between current or prior excessive alcohol use and life expectancy is question that could not usefully be addressed within the available follow-up period of the data in our study and there is no specific literature on alcohol use in HD and effects on life expectancy. In neurodegenerative conditions more generally, the picture is also complex. In dementia and mild cognitive impairment (MCI), a low or moderate alcohol intake is associated with longer life expectancy and has even been speculated to be protective, whereas severe excess is prognostically unfavourable [15,21,22].

There is a significant minority of HD patients who reported a past history of alcohol problems. Within this subset of patients, there was an increased burden of disease. These results provide the clinician with an additional factor to address patients with problem alcohol use; in particular to be aware of more frequent depression, irritability and apathy and the value of counselling pre-manifest carriers of the HD genetic expansion about the risk of high alcohol intake.

## 5. Conclusions

We have demonstrated clear instances of excess alcohol intake affecting well-being in HD, both in a specific patient and within the Enroll-HD participants. Undoubtedly, the combination of substance misuse and of this monogenetic neurodegenerative disease adversely affects patients and their families. In our case report, the patient’s obsession was important in the development of excess alcohol use and in its management, with consequent improvement in clinical symptoms. The patient’s obsessionality was important in the development of excess alcohol use and in its management, with consequent improvement in clinical symptoms. Alcohol excess may be the most readily documented of all the forms of recreational substance misuse harmful to the brain. The Huntington gene expansion is very likely the most common individual genetic abnormality that leads to a dementing illness, and in Enroll-HD is a unique global register for a specific neurodegenerative dementia diagnosis. Despite the reality of substance use as a problem in dementia and our focus on aspects which were relatively well-defined (namely alcohol as a substance of misuse and the Enroll-HD database) the experiences of and ill-effects for individual patients of substance misuse were too varied for specific patterns to be discerned. Alcohol excess and misuse of other substances is likely to be an underestimated cause of harm in HD, in other specific and rare neurodegenerative diseases and in the dementing diseases more generally.

## Figures and Tables

**Figure 1 ijerph-18-05113-f001:**
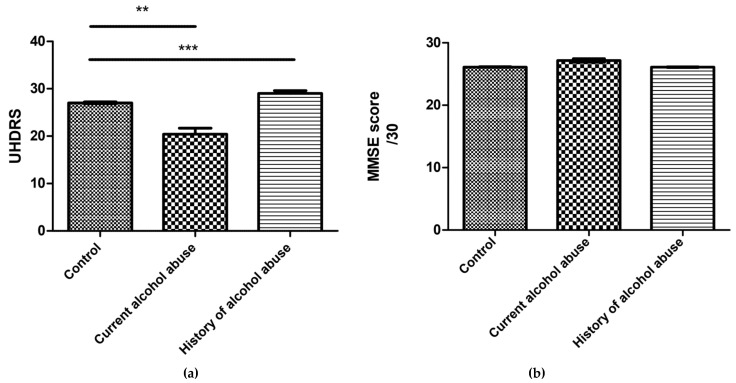
(**a**) Unified Huntington’s Disease Ratings Scale (UHDRS) scores, a measure of motor impairment, are lower in the control patient group, when compared with those who reported alcohol problems in the past (Kruskal–Wallis, post hoc Dunn’s, *p* < 0.0001); (**b**) Mini Mental State Examination (MMSE) when used as a measure of cognition is not related to alcohol or past alcohol use (Kruskal–Wallis, post hoc Dunn’s, *p* > 1). *p* values <0.01 shown as ‘**’ and *p* < 0.001 as ‘***’.

**Figure 2 ijerph-18-05113-f002:**
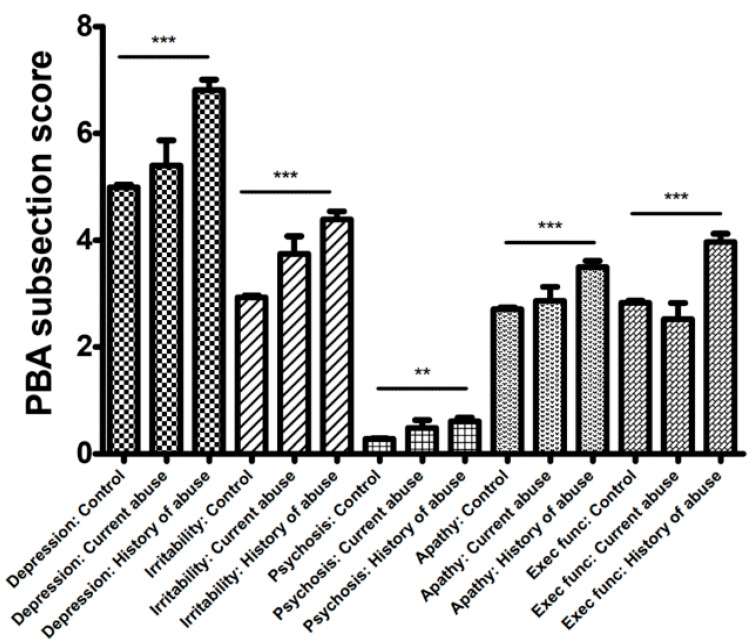
Problem Behaviours Assessments (PBAs) as a measure of psychiatric burden in Huntington’s Disease (HD) shows depression, apathy, irritability, psychosis and executive function are all significantly more sever when compared to controls (Kruskal Wallis, post hoc Dunn’s, *p* < 0.0001). *p* values <0.01 shown as ‘**’ and *p* <0.001 as ‘***’.

## Data Availability

Restrictions apply to the availability of these data. Data was obtained from the Enroll-HD dataset and are available from Enroll-HD.

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
