# Peer review of "Genetic and Environmental Contributors to Neurodegeneration: An Exploration of the Effects of Alcohol on Clinical Features of Huntington’s Disease Using the Enroll-HD Global Platform"

_ijerph, 2021, doi:10.3390/ijerph18105113_

Round 1

Reviewer 1 Report

Symonds et. al present a correlation of Huntington's disease (HD) and alcohol consumption in their manuscript that is based on a dataset of more than sixteen thousand patients that have HD. The evaluation of the Unified Huntington's Disease Ratings Scale and the Problem Behaviours Assessment of the patient cohort is the core of this study presented in figure 1. The presented data puts alcohol abuse in a relevant context with the motor features and psychiatric burden of HD. 
The choice of the case vignette of the authors is a bit ambiguous, as the patient new of her HD prognosis and then before turning to to use alcohol habitually. This can lead to the conclusion that this patient is using alcohol as a remedy of choice to cope with her HD prognosis and symptoms. 

Author Response

Thank you for this constructive comment.

We agree that there is a possibility that the patient consumed alcohol in excess as a reaction to her condition. We have now acknowledged this in the text. This point was omitted in the original text as our focus was on the way in which alcohol misuse was managed in this patient, rather than its complex causation.

Reviewer 2 Report

This study needs specific follow up - in the case study: is that typical behaviour ? -as neurodegeneration occurs does it lead to substance misuse- if so this can be prevented in those who have the dysfunctional gene. 

Author Response

Thank you for this constructive comment.

We agree that the effects of alcohol on longevity in HD is an interesting point but it is little studied and was not something that could be commented upon in the study as insufficient duration of follow-up data was available. We resorted instead to comparison of the available data on ‘past alcohol use’ versus ‘present’ use.

Reviewer 3 Report

In this paper, authors investigated the data of 16120 HD patients in 21 countries. Alcohol problems were more common in the group of manifest HD than pre-manifest HD. The severity of clinical impairments in HD was influenced by alcohol. High alcohol intake had a significant higher motor impairments, but not lower MMSE score.

An analysis in HD with a huge number of 16120 people provides crucial conclusions and is worth of acceptance after some revisions.

The interpretation and limitations of the results of this paper are carefully described in the discussion.

The reviewer has some minor concerns.

1) Have you analyzed the relationship between alcohol intake and life prognosis? This information is important for the treatment of HD patients.

2) Did you analyze the relationship between MMSE, CAG repeat, and life prognosis? The reviewer would like to know the consistency between the past researches and the results of this paper.

3) How about adding some discussion on the relationship between alcohol intake and other neurodegenerative diseases other than HD?

Author Response

Thank you for these constructive comments.

  1. We agree that the effects of alcohol on longevity in HD is an interesting point but it is little studied and was not something that could be commented upon in the study as insufficient duration of follow-up data was available. We resorted instead to comparison of the available data on ‘past alcohol use’ versus ‘present’ use. However, we agree that future studies could look at the relationship of alcohol uses and surrogate measures of prognosis in Enroll-HD such as total functional score and ADL as well as mortality; one problem is that such data may be skewed in that participants tend to drop off the study towards the advanced stages of the disease.
  2. We have added a comment on this point to the text. The listed factors (Cognitive function and MMSE) do affect prognosis, independently of alcohol use in HD but, in the interests of keeping the paper concise, we have not discussed at length. Preliminary analysis of relationships between these important factors found that the EnrollHD dataset did not permit any clear conclusions. As described in the introduction, the prior data on alcohol use and harm in HD are equivocal.
  3. This is an important question, especially as the journal’s issue concerns dementia and neurodegeneration in general, rather than HD specifically. Alcohol undoubtedly contributes to brain disease/degeneration and complicates the course of several neurodegenerative diseases. We have added to the text to describe this. However, the nature of the data (the lack of prospective studies, the difficult of controlling for various lifestyle factors which may differ greatly in those drinking conventionally moderate quantities of alcohol and those of drink somewhat greater quantities) is such that conclusions are problematic. We believe that some useful inferences have been drawn from analysis of a very specific neurodegenerative diagnosis (HD) but accept that the dataset available did not afford opportunities for definitive conclusions. Further, prospective studies would be required for such conclusions but these would be ethically (and logistically) challenging.